# Eye of Horus: A Vision-based Framework for Real-time Water Level Measurement

Mohammad H. Erfani[1], Corinne Smith[2], Zhenyao Wu[3], Elyas Asadi Shamsabadi[4],
Farboud Khatami[1], Austin R.J. Downey[1,2], Jasim Imran[1], and Erfan Goharian[*1]

[1]Department of Civil & Environmental Engineering, University of South Carolina Columbia, SC 29208, USA
[2]Department of Mechanical Engineering, University of South Carolina, Columbia, SC 29208, USA
[3]Department of Computer Science & Engineering, University of South Carolina, Columbia, SC 29201, USA
[4]School of Civil Engineering, Faculty of Engineering, The University of Sydney, Sydney, NSW 2006, Australia

## Abstract

Heavy rains and tropical storms often result in floods, which are expected to increase in frequency and intensity. Flood prediction models and inundation mapping tools provide decision-makers and emergency responders with crucial information to better prepare for these events. However, the performance of models relies on the accuracy and timeliness of data received from in-situ gaging stations and remote sensing; each of these data sources has its limitations, especially when it comes to real-time monitoring of floods. This study presents a vision-based framework for measuring water levels and detecting floods using Computer Vision and Deep Learning (DL) techniques. The DL models use time-lapse images captured by surveillance cameras during storm events for the semantic segmentation of water extent in images. Three different DL-based approaches, namely PSPNet, TransUNet, and SegFormer, were applied and evaluated for semantic segmentation. The predicted masks are transformed into water level values by intersecting the extracted water edges, with the 2D representation of a point cloud generated by an Apple iPhone 13 Pro LiDAR sensor. The estimated water levels were compared to reference data collected by an ultrasonic sensor. The results showed that SegFormer outperformed other DL-based approaches by achieving 99.55% and 99.81% for Intersection over Union (IoU) and accuracy, respectively. Moreover, the highest correlations between reference data and the vision-based approach reached above 0.98 for both the coefficient of determination ($r^2$) and Nash-Sutcliffe Efficiency. This study demonstrates the potential of using surveillance cameras and Artificial Intelligence for hydrologic monitoring and their integration with existing surveillance infrastructure.

## 1   Introduction

Flood forecasts and Flood Inundation Mapping (FIM) can play an important role in saving human lives and reducing damages by providing timely information for evacuation planning, emergency management, and relief efforts [Gebrehiwot et al., 2019]. These models and tools are designed to identify and predict inundation areas and the severity of damage caused by storm events. Two primary sources of data for these models are in-situ gaging networks and remote sensing. For example, in-situ stream gages, such as those operated by the United States Geological Survey (USGS) provide useful streamflow information like water height and discharge at monitoring sites [Turnipseed and Sauer, 2010]. However, they cannot provide an adequate spatial resolution of streamflow characteristics [Lo et al., 2015]. The limitation of in-situ stream gages is further exacerbated by the lack of systematic installation along the waterways and accessibility issues [Li et al., 2018; King et al., 2018]. Satellite data and remote sensing can complement in-situ gage data by providing information at a larger spatial scale [Alsdorf et al., 2007]. However, continuous monitoring data for a region of interest remains to be a problem due to the limited revisit intervals of satellites, cloud cover, and systematic departures or biases [Panteras and Cervone, 2018]. Crowdsourcing methods have gained attention as a potential solution but their reliability is questionable [Schnebele et al., 2014; Goodchild, 2007; Howe, 2008]. To address these limitations and enhance real-time monitoring capabilities, surveillance cameras are inves-

---

[*]goharian@cec.sc.edu

tigated here as a new source of data for hydrologic monitoring and flood data collection. However, this requires a significant investment in Computer Vision (CV) and Artificial Intelligence (AI) techniques to develop reliable methods for detecting water in surveillance images and translating that information into numerical data.

Recent advances in CV offer new techniques for processing image data for the quantitative measurements of physical attributes from a site [Forsyth and Ponce, 2002]. However, there is limited knowledge of how visual information can be used to estimate physical water parameters using CV techniques. Inspired by the principle of the float method, Tsubaki et al. [2011] used different image processing techniques to analyze images captured by closed-circuit television (CCTV) systems installed for surveillance purposes to measure the flow rate during flood events. In another example, Kim et al. [2011] proposed a method for measuring water level by detecting the borderline between a staff gauge and the surface of water based on image processing of the captured image of the staff gage installed in the middle of the river. As the use of images for environmental monitoring becomes more popular, several studies have investigated the source and magnitude of errors common in image-based measurement systems, such as the effect of image resolution, lighting effects, perspective, lens distortion, water meniscus, and temperature changes [Elias et al., 2020; Gilmore et al., 2013]. Furthermore, proposed solutions to resolve difficulties originating from poor visibility have been developed to better identify readings on staff gages [Zhang et al., 2019]. Recently, Deep Learning (DL) has become prevalent across a wide range of disciplines, particularly in applied sciences such as CV and engineering.

DL-based models have been utilized by the water resources community to determine the extent of water and waterbodies visible in images captured by surveillance camera systems. These models can estimate the water level [Pally and Samadi, 2022]. In a similar vein, Moy de Vitry et al. [2019]; Vandaele et al. [2021] employed a DL-based approach to identify floodwater in surveillance footage and introduced a novel qualitative flood index, SOFI, to determine water level fluctuations. SOFI was calculated by taking the aspect ratio of the area of the water surface detected within an image to the total area of the image. However, these types of methods, which make prior assumptions and estimate water level fluctuation roughly, cannot serve as a vision-based alternative for measuring streamflow characteristics. More systematic studies adopted photogrammetry to reconstruct a high-quality 3D model of the environment with a high spatial resolution to have a precise estimation of real-world coordination while measuring streamflow rate and stage. For example, Eltner et al. [2018, 2021] introduced a method based on Structure from Motion (SfM), and photogrammetric techniques, to automatically measure the water stage using low-cost camera setups.

Advances in photogrammetry techniques enable 3D surface reconstruction with a high temporal and spatial resolution. These techniques are adopted to build 3D surface models from RGB imagery [Westoby et al., 2012; Eltner and Schneider, 2015; Eltner et al., 2016]. However, most of the photogrammetric methods are still expensive as they rely on differential global navigation satellite systems (DGNSS), ground control points (GCPs), commercial software, and data processing on an external computing device [Froideval et al., 2019]. A LiDAR scanner, on the other hand, is now easily available since the introduction of the iPad Pro and iPhone 12 Pro in 2020 by Apple. This device is the first smartphone equipped with a native LiDAR scanner and offers a potential paradigm shift in digital field data acquisition which puts these devices at the forefront of smartphone-assisted fieldwork [Tavani et al., 2022]. So far, the iPhone LiDAR sensor has been used in different studies such as forest inventories [Gollob et al., 2021] and coastal cliff site [Luetzenburg et al., 2021]. The availability of LiDAR sensors to build 3D environments, and advancements in DL-based models offer a great potential to produce numerical information from ground-based imageries.

This paper presents a vision-based framework for measuring water levels from time-lapse images. The proposed framework introduces a novel approach by utilizing the iPhone LiDAR sensor as a laser scanner, which is commonly available on consumer-grade devices, for scanning and constructing a 3D point cloud of the region of interest. During the data collection phase, time-lapse images and ground truth water level values were collected using an embedded camera and ultrasonic sensor. The water extent in the captured images was determined automatically using semantic segmentation DL-based models. For the first time, the performance of three different state-of-the-art DL-based approaches, including Convolutional Neural Networks (CNN), hybrid CNN-Transformer, and Transformers-Multilayer Perceptron (MLP), was evaluated and compared. CV techniques were applied for camera calibration, pose

estimation of the camera setup in each deployment, and 3D-2D reprojection of the point cloud onto the image plane. Finally, K-Nearest Neighbors (KNN) was used to find the nearest projected (2D) point cloud coordinates to the water line on the river banks, for estimating the water level in each time-lapse image.

# 2  Deep Learning Architectures

Since this study tends to cover a wide range of DL approaches, this section solely focuses on reviewing different DL-based architectures. So far, different DL networks were applied and evaluated for semantic segmentation of the waterbodies within the RGB images captured by cameras [Erfani et al., 2022]. All existing semantic segmentation approaches–CNN and Transformer-based– share the same objective of classifying each pixel of a given image but differ in the network design.

CNN-based models were designed to imitate the recognition system of primates [Shamsabadi et al., 2022], while possessing different network designs such as low-resolution representations learning [Long et al., 2015; Chen et al., 2017], high-resolution representations recovering [Badrinarayanan et al., 2015; Noh et al., 2015; Lin et al., 2017], contextual aggregation schemes [Yuan and Wang, 2018; Zhao et al., 2017; Yuan et al., 2020], feature fusion and refinement strategy [Lin et al., 2017; Huang et al., 2019; Li et al., 2019; Zhu et al., 2019; Fu et al., 2019]. CNN-based models follow local to global features in different layers of the forward pass, which used to be thought of as a general intuition of the human recognition system. In this system, objects are recognized through the analysis of texture and shape-based clues– local and global representations and their relationship in the entire field of view. Recent research, however, shows significant differences exist between the visual behavioral system of humans and CNN-based models [Geirhos et al., 2018b; Dodge and Karam, 2017; De Cesarei et al., 2021; Geirhos et al., 2020, 2018a], and reveal higher sensitivity of the visual systems in humans to global features rather than local ones [Zheng et al., 2018]. This fact drew attention to models that focus on the global context in their architectures.

Developed by Dosovitskiy et al. [2020], Vision Transformer (ViT) was the first model that showed promising results on a computer vision task (image classification) without using convolution operation in its architecture. In fact, ViT adopts "Transformers," as a self-attention mechanism, to improve accuracy. "Transformer" was initially introduced for sequence-to-sequence tasks such as text translation [Vaswani et al., 2017]. However, as applying the self-attention mechanism on all image pixels is computationally expensive, the Transformer-based models could not compete with the CNN-based models until the introduction of ViT architecture which applies self-attention calculations on the low-dimension embedding of small patches originating from splitting the input image, to extract global contextual information. Successful performance of ViT on image classification inspired several subsequent works on Transformer-based models for different computer vision tasks [Liu et al., 2021].

In this study, three different DL-based approaches including CNN, hybrid CNN-Transformer, and Transformers-Multilayer Perceptron (MLP) were trained and tested for semantic segmentation of water. For these approaches, the selected models were PSPNet [Zhao et al., 2017], TransUNet [Chen et al., 2021] and SegFormer [Xie et al., 2021], respectively. The performance of these models is evaluated and compared using conventional metrics, including class-wise Intersection over Union (IoU) and per-pixel accuracy (ACC).

# 3  Study Area

In order to evaluate the performance of the proposed framework for measuring the water levels in rivers and channels, a time-lapse camera system has been deployed at Rocky Branch, South Carolina. This creek is approximately 6.5 km long and collects stormwater from the University of South Carolina campus and the City of Columbia. Rocky Branch is subjected to rapid changes in water flow and discharges into the Congaree River [Morsy et al., 2016]. The observation site is located within the University of South Carolina campus behind 300 Main Street (see Figure 1a).

An Apple iPhone 13 Pro LiDAR sensor was used to scan the region of interest. Although there is no official information about the technology and hardware specifications, Gollob et al. [2021] reports

the LiDAR module operates at the 8XX nm wavelength and consists of an emitter (Vertical Cavity Surface-Emitting Laser with Diffraction Optics Element, VCSEL DOE) and a receptor (Single Photon Avalanche Diode array-based Near Infrared Complementary Metal Oxide Semiconductor image sensor, SPAD NIR CMOS) based on direct-time-of-flight technology. Comparisons between the Apple LiDAR sensor and other types of laser scanners including hand-held, industrial, and terrestrial have been conducted by several recent studies [Mokroš et al., 2021; Vogt et al., 2021]. Gollob et al. [2021] tested and reported the performance of a set of eight different scanning apps, and found three applications including 3D Scanner App, Polycam and SiteScape suitable for actual practice tests. The objective of this study is not the evaluation of the iPhone LiDAR sensor and app performance. Therefore, the 3D Scanner App [LABS, 2022] was used with the following settings: confidence = high, range = 5.0 m, masking = None, and resolution = 5 mm, for scanning and 3D reconstruction processing. The scanned 3D point cloud and its corresponding scalar field are shown in Figure 1b and Figure 1c, respectively.

As the LiDAR scanner settings were set at the highest level of accuracy and computational demand, scanning the whole region of interest at the same time was not possible. So, the experimental region was divided into several sub-regions and scanned in multi-step. In order to assemble the sub-region LiDAR scans, several GCPs were considered in the study area. These GCPs were measured by a total station (Topcon GM Series) and used as landmarks to align distinct 3D point clouds with each other and create an integrated point cloud encompassing the entirety of the study area.

Moreover, several ArUco markers were installed for estimating camera (extrinsic) parameters. In each setup deployment, these parameters should be recalculated (additional information can be found in section 4.3). Since it was not possible to accurately measure the real-world coordination of ArUco markers by the LiDAR scanner, the coordinates of the top-left corner of markers were also measured by the surveying total station. To establish a consistent coordinate system, the 3D point cloud scanned for each sub-region was transformed into the total station's coordinate system. The real-world coordinates of ArUco markers were then added to the 3D point cloud (see Figure 1b).

# 4 Methodology

This study introduces the Eye of Horus, a vision-based framework for hydrologic monitoring and real-time water level measurements in bodies of water. The proposed framework includes three main components. The first step is designing two deployable setups for data collection. These setups consist of a programmable time-lapse camera run by Raspberry Pi and an ultrasonic sensor run by Arduino. After collecting data, the first phase (Module 1) involves configuring and training DL-based models for semantic segmentation of water in the captured images. In the second phase (Module 2), CV techniques for camera calibration, spatial resection, and calculating projection matrix are discussed. Finally, in the third phase (Module 3), an ML-based model uses the information achieved by CV models to find the relationships between real-world coordinates of water level in the captured images (see Figure 2).

## 4.1 Data Acquisition

Two different single-board computers (SBC) were used in this study, Raspberry Pi (Zero W) for capturing time-lapse images of a river scene, and Arduino (Nano 3.x) for measuring water level as the ground truth data. These devices were designed to communicate with each other, i.e., to trigger the other to start or stop recording. During capturing time-lapse images, the Pi camera device triggers the ultrasonic sensor for measuring the corresponding water level. The camera device is equipped with the Raspberry Pi Camera Module 2 which has a Sony IMX219 8-megapixel sensor. This sensor is able to capture an image size of $4,256 \times 2,832$ pixels. However, in this study, the image resolution was set to $1,920 \times 1,440$ pixels to balance image quality and computational cost in subsequent image processing steps. This setup is also equipped with a 1200 mAh UPS lithium battery power module to provide uninterrupted power to the Pi SBC (see Figure 3a).

The Arduino-based device records the water level. The design is based on an unmanned aerial vehicle (UAV) deployable sensor created by Smith et al. [2022]. The nRF24L01+ single-chip 2.4 GHz transceiver allows the Arduino and Raspberry Pi to communicate via radio frequency (RF). The chip

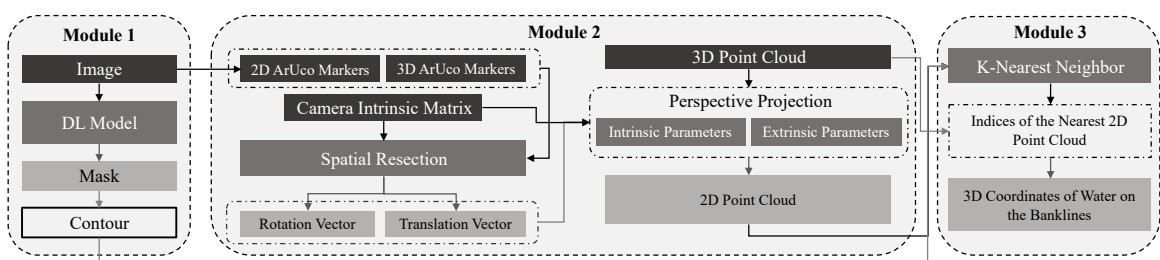

(a)

(b)

(c)

Figure 1: Study area of the Rocky Branch Creek. (a) View of the region of interest, (b) The scanned 3D point cloud of the region of interest including an indication of the ArUco markers' locations, and (c) The scalar field of left and right banks of Rocky Branch in the region of interest (the colorbar and the frequency distribution of $z$ values for the captured points are shown on the right side).

Figure 2: The Eye of Horus workflow includes three main modules starting from processing images captured by the time-lapse camera to estimating water level by projecting the waterline on river banks using CV techniques.

is housed in both packages and the channel, pipe addresses, data rate, and transceiver/receiver configuration are all set in the software. The HC-SR04 ultrasonic sensor is mounted to the base of the Arduino device and provides a contactless water level measurement. Two permanent magnets at the top of the housing attach to a ferrous structure and allow the ultrasonic sensor to be suspended up to 14 feet over the surface of the water. The device also includes a microSD card module and DS3231 real-time clock, which enable data logging and storage on-device as well as transmission. The device is powered by a rechargeable 7.4V 1500 mAh lithium polymer battery (see Figure 3b).

The Arduino device waits to receive a ping from the Raspberry Pi device to initiate data collection. The ultrasonic sensor measures the distance from the sensor transducer to the surface of the water. The nRF24L01+ transmits this distance to the Raspberry Pi device and saves the measurement and a time stamp from the real-time clock to an onboard microSD card. This acts as backup data storage, in case transmission to the Raspberry Pi fails. The nRF24L01+ RF transceivers have an experimentally determined range of up to 30 ft which allows flexibility in the relative placement of the camera to the measuring site.

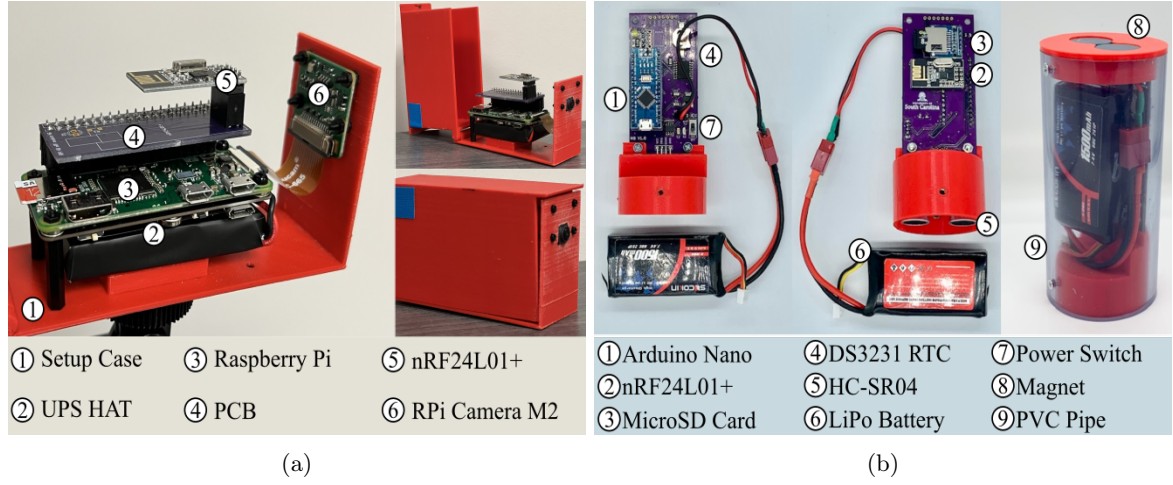

| ① Setup Case | ③ Raspberry Pi | ⑤ nRF24L01+ |
| --- | --- | --- |
| ② UPS HAT | ④ PCB | ⑥ RPi Camera M2 |

| ①Arduino Nano | ④DS3231 RTC | ⑦Power Switch |
| --- | --- | --- |
| ②nRF24L01+ | ⑤HC-SR04 | ⑧Magnet |
| ③MicroSD Card | ⑥LiPo Battery | ⑨PVC Pipe |

(a)                                      (b)

Figure 3: Data acquisition devices. (a) Beena, run by Raspberry Pi (Zero W) for capturing time-lapse images of the river scene; and (b) Aava, run by Arduino Nano for measuring water level correspondence.

A dataset for semantic segmentation was created by collecting images from a specific region of interest at different times of the day and under various flow regimes. This dataset includes 1,172 images, with manual annotations of the streamflow in the creek for all of them. The dataset is further divided into 812 training images, 124 validation images, and 236 testing images.

## 4.2   Deep Learning Model for Water Segmentation

The water extent can be automatically determined on the 2D image plane with the help of DL-based models. The task of semantic segmentation was applied within the framework of this study to delineate the water line on the left and right banks of the channel. Three different DL-based models were trained and tested in this study. PSPNet, the first model, is a CNN-based semantic segmentation multi-scale network which can better learn the global context representation of a scene [Zhao et al., 2017]. ResNet-101 [He et al., 2016] was used as the backbone of this model to encode input images into the features. ResNet architecture takes the advantage of "Residual blocks" that assist the flow of gradients during the training stage allowing effective training of deep models even up to hundreds of layers. These extracted features are then fed into a pyramid pooling module in which feature maps produced by small to large kernels are concatenated to distinguish patterns of different scales [Minaee et al., 2021].

TransUNet, the second model, is a U-shaped architecture that employs a hybrid of CNN and Transformers as the encoder to leverage both the local and global contexts for precise localization and pixel-wise classification [Chen et al., 2021]. In the encoder part of the network, CNN is first used as a feature extractor to generate a feature map for the input image, which is then fed into Transformers to extract long-range dependencies. The resulting features are upsampled in the decoding path and

combined with detailed high-resolution spatial information skipped from the CNN to make estimations on each pixel of the input image.

SegFormer, the third model, unifies a novel hierarchical Transformer, which does not require the positional encodings used in standard Transformers, and MultiLayer Perceptron (MLP) performs efficient segmentation [Xie et al., 2021]. The hierarchical Transformer introduced in the encoder of this architecture gives the model the attention ability to multiscale features (high-resolution fine and low-resolution coarse information) in the spatial input without the need for positional encodings that may adversely affect a models performance when testing on a different resolution from training. Moreover, unlike other segmentation models that typically use deconvolutions in the decoder path, a lightweight MLP is employed as the decoder of this network that inputs the features extracted at different stages of the encoder to generate a prediction map faster and more efficiently. Two different variants, including SegFormer-B0 and SegFormer-B5, were applied in this study. The configuration of the models implemented in this study is elaborated in Table 1. The total number of parameters (Params), occupied memory size on GPU (Total Size), and input image size (Batch Size) are reported in Million (M), Megabyte (MB), and Batch size×Height×Width×Channel (B, H, W, C) respectively.

Table 1: The configuration of models trained and tested in this study.

| Model Names | Params (M) | Total Size (MB) | Batch Size (B, H, W, C) | Loss Function | Optimizer | LR |
|---|---|---|---|---|---|---|
| PSPNet | 66.2 | 7,178 | $2\times500\times500\times3$ | Binary Cross Entropy | SGD | 2.50E-04 |
| TransUNet | 20.1 | 6,017 | $2\times448\times448\times3$ | Cross Entropy + Dice | SGD | 2.50E-04 |
| SegFormer-B0 | 3.7 | 2,217 | $2\times512\times512\times3$ | Cross Entropy | AdamW | 6.00E-05 |
| SegFormer-B5 | 82.0 | 27,666 | $2\times1024\times1024\times3$ | Cross Entropy | AdamW | 6.00E-05 |

The models were implemented using PyTorch. During the training procedure, the loss function, optimizer, and learning rate were set individually for each model based on the results of preliminary runs used to find the optimal hyperparameters. In the case of PSPNet and TransUNet, the base learning rate was set to $2.5\times10^{-4}$ and decayed using the poly policy [Zhao et al., 2017]. These networks were optimized using stochastic gradient descent (SGD) with a momentum of 0.9 and weight decay of 0.0001. For SegFormer (B0 and B5), a constant learning rate of $6.0\times10^{-5}$ was used, and the networks were trained with the AdamW optimizer [Loshchilov and Hutter, 2017]. All networks were trained for 30 epochs with a batch size of two. The training data for PSPNet and TransUNet were augmented with horizontal flipping, random scaling, and random cropping.

## 4.3   Projective Geometry

In this study, CV techniques are used for different purposes. First, CV models were used for camera calibration. They include focal length, optical center, radial distortion, camera rotation, and translation. These parameters provide the information (parameters or coefficients) about the camera that is required to determine the relationship between 3D object points in the real-world coordinate system and its corresponding 2D projection (pixel) in the image captured by that calibrated camera. Generally, camera calibration models estimate two kinds of parameters. First, the intrinsic parameters of the camera (e.g., focal length, optical center, and radial distortion coefficients of the lens). Second, extrinsic parameters (refer to the orientation– rotation, and translation– of the camera) with respect to the real-world coordinate system.

To estimate the camera intrinsic parameters, OpenCV built-in was applied for camera calibration using a 2D checkerboard [Bradski, 2000]. The focal length ($f_x$, $f_y$), optical centers ($c_x$, $c_y$), and the skew coefficient ($s$) can be used to create a camera intrinsic matrix $\mathbf{K}$:

$$\mathbf{K} = \begin{bmatrix} f_x & s & c_x \\ 0 & f_y & c_y \\ 0 & 0 & 1 \end{bmatrix} \tag{1}$$

The camera extrinsic parameters were determined using the pose computation problem, Perspective-n-Point (PnP), which consists of solving for the rotation, and translation that minimizes the reprojection

error from 2D-3D point correspondences [Marchand et al., 2015]. The PnP estimates the extrinsic parameters given a set of 'object points,' their corresponding 'image projections,' as well as the camera intrinsic matrix and the distortion coefficients. The camera extrinsic parameters can be represented as a combination of a 3×3 rotation matrix $\mathbf{R}$ and a 3×1 translation vector $\mathbf{t}$:

$$[\mathbf{R} \mid \mathbf{t}] = \begin{bmatrix} r_{11} & r_{12} & r_{13} & t_x \\ r_{21} & r_{22} & r_{23} & t_y \\ r_{31} & r_{32} & r_{33} & t_z \end{bmatrix} \tag{2}$$

Equation 3 represents the 'Projection Matrix,' in a homogeneous coordinate system. The projection matrix consists of two parts: the intrinsic matrix ($\mathbf{K}$), containing intrinsic parameters, and the extrinsic matrix ($[\mathbf{R} \mid \mathbf{t}]$) which can be represented as follows:

$$\begin{bmatrix} u \\ v \\ 1 \end{bmatrix} = \overbrace{\begin{bmatrix} f_x & s & c_x & 0 \\ 0 & f_y & c_y & 0 \\ 0 & 0 & 1 & 0 \end{bmatrix}}^{\mathbf{K}} \overbrace{\begin{bmatrix} r_{11} & r_{12} & r_{13} & t_x \\ r_{21} & r_{22} & r_{23} & t_y \\ r_{31} & r_{32} & r_{33} & t_z \\ 0 & 0 & 0 & 1 \end{bmatrix}}^{[\mathbf{R}|\mathbf{t}]} \begin{bmatrix} X_w \\ Y_w \\ Z_w \\ 1 \end{bmatrix} \tag{3}$$

Direct Linear Transformation (DLT) is a mathematical technique commonly used to estimate the parameters of the Projection Matrix. The DLT method requires a minimum of six pairs of known 3D-2D correspondences to establish twelve equations and estimate all parameters of the Projection Matrix. Generally, the intrinsic parameters remain constant for a specific camera model, such as the Raspberry Pi Camera Module 2, and can be reused for all images captured by that camera. However, the extrinsic parameters change whenever the camera's location is altered. Consequently, for each setup deployment, recalculation of the extrinsic parameters is necessary to reconstruct the Projection Matrix. To simplify this process, the PnP method was replaced with DLT. It can reduce the required number of 3D-2D correspondence pairs to three, by reusing the intrinsic parameters.

Additionally, ArUco markers were incorporated to represent pairs of known 3D-2D correspondences. For this purpose, the pixel coordinates of ArUco markers were determined using the OpenCV ArUco marker detection module on the 2D image plane, and the corresponding 3D real-world coordinates were measured by the total station. With these 3D-2D point correspondences, the spatial position and orientation of the camera can be estimated for each setup deployment. After retrieving all the necessary parameters, a full-perspective camera model can be generated. Using this model, the 3D point cloud is projected onto the 2D image plane. The projected (2D) point cloud represents the 3D real-world coordinates of the nearest 2D pixel correspondence on the image plane

## 4.4   Machine Learning for Image Measurements

Using the projection matrix, the 3D point cloud is projected on the 2D image plane (see Figure 4). The projected (2D) point cloud is intersected with the water line pixels, the output of the DL-based model (Module 1), to find the nearest point cloud coordinate. To achieve this objective, we utilize the K-Nearest Neighbors (KNN) algorithm. Notably, the indices of the selected points remain consistent for both the 3D point cloud and the projected (2D) correspondences. As a result, by utilizing the indices of the chosen projected (2D) points, the corresponding real-world 3D coordinates can be retrieved.

## 4.5   Performance Metrics

The performance of the proposed framework is evaluated based on four different metrics including coefficient of determination ($r^2$), Nash-Sutcliffe Efficiency (NSE), Root Mean Square Error (RMSE), and Percent bias (PBIAS). $R^2$ is a widely used metric that quantifies how much of the observed dispersion can be explained in a linear relationship by the prediction.

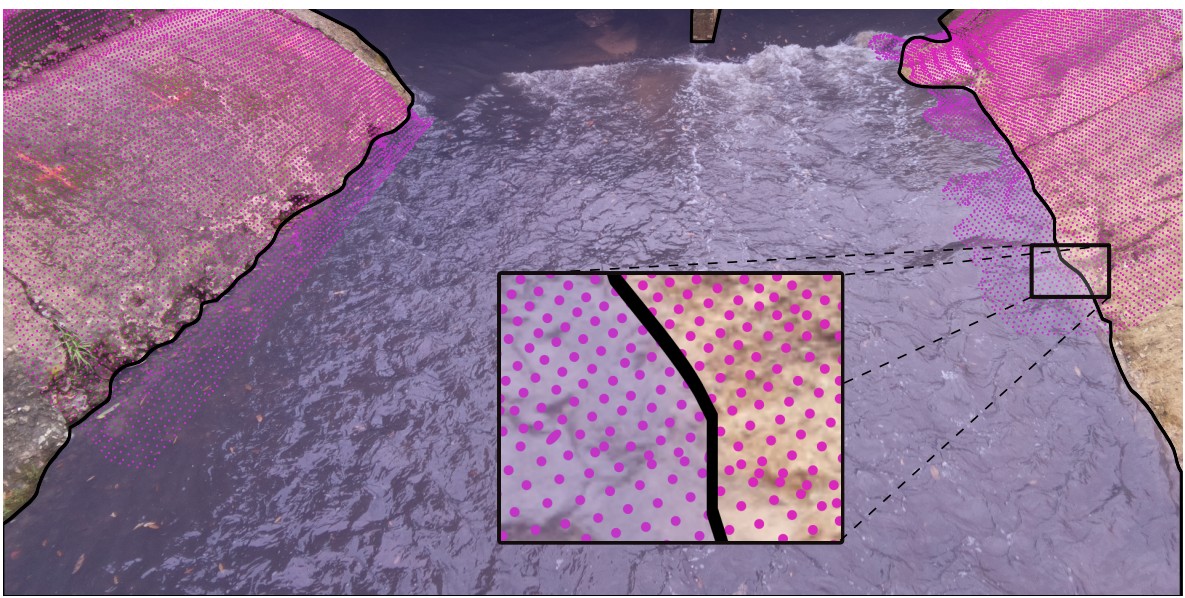

Figure 4: KNN is used to find the nearest projected (2D) point cloud (magenta dots) to the water line (black line) on the image plane.

$$r^2 = \left( \frac{\sum_{i=1}^{n} (O_i - \bar{O})(P_i - \bar{P})}{\sqrt{\sum_{i=1}^{n} (O_i - \bar{O})^2 \cdot \sum_{i=1}^{n} (P_i - \bar{P})^2}} \right)^2 \qquad (4)$$

However, if the model systematically over- or under-estimates the results, $r^2$ will still be close to 1.0
as it only takes dispersion into account [Krause et al., 2005]. NSE, another commonly used metric
in hydrology, presents the model performance with an interpretable scale and is used to differentiate
between 'good' and 'bad' models [Knoben et al., 2019].

$$NSE = 1 - \frac{\sum_{i=1}^{n}(O_i - P_i)^2}{\sum_{i=1}^{n}(O_i - \bar{O})^2} \qquad (5)$$

RMSE represents the square root of the average of squares of the errors, the differences between
predicted values and observed values.

$$RMSE = \sqrt{\frac{1}{n} \sum_{i=1}^{n}(O_i - P_i)^2} \qquad (6)$$

The PBIAS of estimated water level, compared against the ultrasonic sensor data was also used to
show where the two estimates are close to each other and where they significantly diverge [Lin et al.,
2020].

$$PBIAS = \frac{100}{n} \sum_{i=1}^{n} \frac{(O_i - P_i)}{\sum_{i=1}^{n} O_i} \qquad (7)$$

Where $n$ is the number of data points, $O$ and $P$ are observed and predicted values, respectively.

# 5  Results and Discussion

The results of this study are presented in two sections. First, the performance of DL-based models is discussed. Then, in the second section, the performance of the proposed framework is evaluated for five different deployments.

## 5.1  DL-based Models Results

The performance of DL-based models for the task of semantic segmentation is evaluated and compared in this section. Since the proposed dataset includes just two classes, "river" and "non-river", "non-river" was omitted from the evaluation process, and the performance of models is only reported for the "river" class of the test set. The class-wise intersection over union (IoU) and the per-pixel accuracy (ACC) were considered the main evaluation metrics in this study. According to Table 2, both variants of SegFormer– SegFormer-B0, and SegFormer-B5– outperform other semantic segmentation networks on the test set. Considering the models' configurations detailed in Table 1, SegFormer-B0 can be considered the most efficient DL-based network, as it is comprised of only 3.7 M trainable parameters and occupies just 2,217 Megabytes of GPU ram during training. In Figure 5, four different visual representations of the models' performance on the validation set of the proposed dataset are presented. Since the water level is estimated by intersecting the water line on river banks with the projected (2D) point cloud, precise delineation of the water line is of utmost importance to achieve better results in the following steps. This means that estimating the correct location of the water line on creek banks in each time-lapse image plays a more significant role than performance metrics in this study. Taking the quality of water line detection into account and based on the visual representations shown in Figure 5, SegFormers' variants still outperform DL-based approaches. In this regard, a comparison of PSPNet and TransUNet showed that PSPNet can delineate the water line more clearly, while the segmented area is more integrated for TransUNet outputs.

Table 2: The performance metrics of different DL-based approaches.

| Model Names | IoU (River) | ACC (River) |
|---|---|---|
| PSPNet | 94.88% | 95.84% |
| TransUNet | 93.54% | 96.89% |
| SegFormer-B0 | 99.38% | 99.77% |
| SegFormer-B5 | 99.55% | 99.81% |

CNNs are typically limited by the nature of their convolution operations, leading to architecture-specific issues such as locality [Geirhos et al., 2018a]. Consequently, CNN-based models may achieve high accuracy on training data, but their performance can decrease considerably on unseen data. Additionally, compared to Transformer-based networks, they perform poorly at detecting semantics that requires combining long- and short-range dependencies. Transformers can relax the biases of DL-based models inducted by Convolutional operations, achieving higher accuracy in localization of target semantics and pixel-level classification with lower fluctuations in varied situations through the leverage of both local and global cues [Naseer et al., 2021]. Yet, various transformer-based networks may perform differently depending on the targeted task and the network's architecture. TransUNet adopts Transformers as part of its backbone; however, Transformers generate single-scale low-resolution features as output [Xie et al., 2021], which may limit the accuracy when multi-scale objects or single objects with multi-scale features are segmented. The problem of producing single-scale features in standard Transformers is addressed in SegFormer variants through the use of a novel hierarchical Transformer encoder [Xie et al., 2021]. This approach has resulted in human-level accuracy being achieved by Segformer-B0 and -B5 in the delineation of the water line, as shown in Figure 5. The predicted masks are in satisfactory agreement with the manually annotated images.

## 5.2  Water Level Estimation

This section reports the framework performance based on several deployments in the field. The performance results are separately shown for the left and right banks and compared with ultrasonic sensor data as the ground truth. The ultrasonic sensor was evaluated previously that documented an average

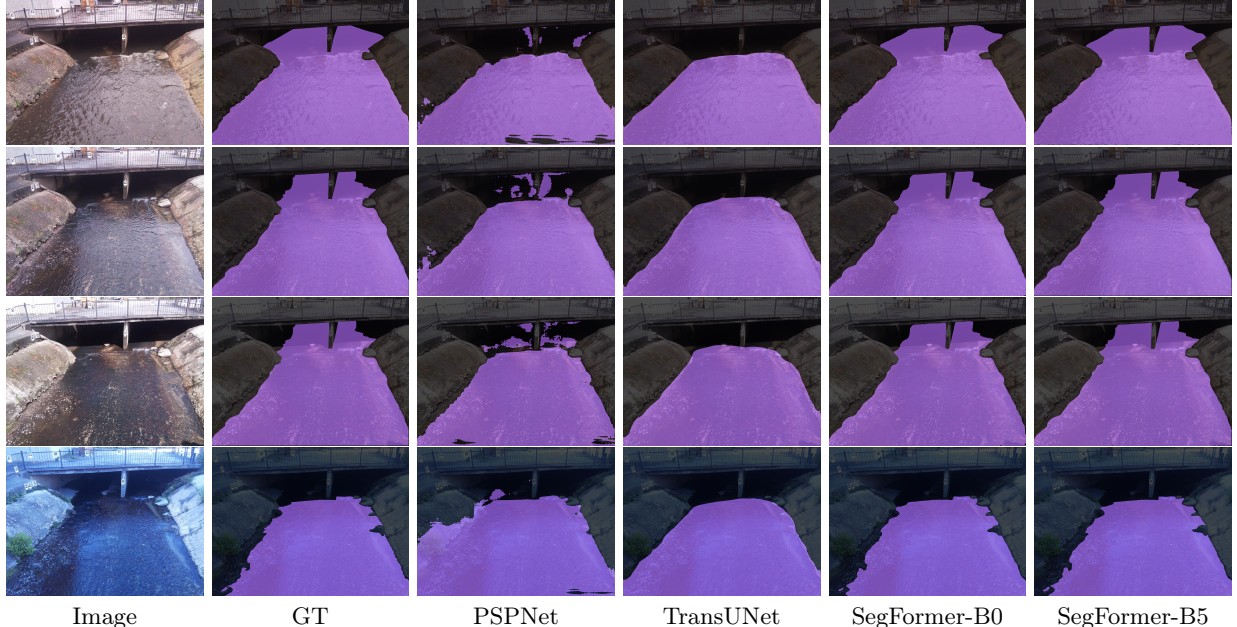

|       |      |         |           |              |              |
|-------|------|---------|-----------|--------------|--------------|
| Image | GT   | PSPNet  | TransUNet | SegFormer-B0 | SegFormer-B5 |

Figure 5: Visual representations of different DL-based image segmentation approaches on the validation dataset.

distance error of 6.9 mm [Smith et al., 2022]. The setup was deployed on several rainy days. The results of each deployment are reported in Table 3.

Table 3: The performance metrics of the framework for five different days of setup deployment.

| Deployment Date | Position       | Metrics |        |        |          |
|-----------------|----------------|---------|--------|--------|----------|
|                 |                | $r^2$   | NSE    | RMSE   | PBIAS    |
| Aug/17/2022     | Left Bankline  | 0.8019  | 0.5258 | 0.0409 | 10.6401  |
|                 | Right Bankline | 0.7932  | 0.7541 | 0.0294 | -0.4848  |
| Aug/19/2022     | Left Bankline  | 0.7701  | 0.5713 | 0.0647 | 16.1015  |
|                 | Right Bankline | 0.9678  | 0.9588 | 0.0201 | -3.4752  |
| Aug/25/2022     | Left Bankline  | 0.7690  | 0.5700 | 0.0435 | -7.7091  |
|                 | Right Bankline | 0.8922  | 0.8711 | 0.0238 | -1.7738  |
| Nov/10/2022     | Left Bankline  | 0.9461  | 0.8129 | 0.0511 | -13.1183 |
|                 | Right Bankline | 0.9857  | 0.9790 | 0.0171 | -1.5210  |
| Nov/11/2022     | Left Bankline  | 0.9588  | 0.8881 | 0.0397 | -10.3656 |
|                 | Right Bankline | 0.9855  | 0.9829 | 0.0155 | -1.7987  |

In addition to Table 3, the results of each deployment are visually demonstrated in Figure 6. The scatter plots show the relationships between the ground truth data (measured by the ultrasonic sensor), and the banks of the river. The scatter plots visually present whether the camera readings overestimate or underestimate the ground truth data. Moreover, the time-series plot of water level is shown for each deployment separately. A hydrograph, showing changes in the water level of a stream over time can be a useful tool for demonstrating whether camera readings can satisfactorily capture the response of a catchment area to rainfall. The proposed framework can be evaluated in terms of its ability to accurately track and identify important characteristics of a flood wave, such as the rising limb, peak, and recession limb.

The first deployment was done on Aug 17, 2022 (see Figure 6a). The initial water level of the base flow and parts of the rising limb were not captured in this deployment. Table 3 shows that the performance results of the right bank camera readings are better than those of the left bank. $R^2$ for both banks was about 0.80 showing a strongly related correlation between the water level estimated by

the framework and ground truth data. Figure 6a shows how the left and right bank camera readings perform during the rising limb; the right bank camera readings still underestimated the water level during this time frame, and during the recession limb, the left bank camera readings overestimated the water level. However, the hydrograph plot shows that both left and right bank camera readings were able to capture the peak water level.

The second deployment was done on Aug 19, 2022. In this deployment, all segments of the hydrograph were captured. According to Table 3, the performance of the right bank camera readings was better than the left bank one; more than 0.95 was reported for $R^2$ and NSE of the right bankline. Figure 6b shows during the rising limb and crest segment both banks estimated the water level similar to ground truth. During the recession limb, the right bank water level estimation kept coincident with ground truth, while the left bank overestimated the water level. The third deployment was on Aug 25, 2022. This time water level of the recession limb and the following base flow were captured (see Figure 6c). The right bank camera readings with $R^2$ of 0.89 performed better than the left bank. This time, left bank camera readings underestimated the water level over the recession limb, but during the following base flow, the water level was estimated correctly by cameras on both banks.

The results indicate that the right bank camera readings performed better than the left bank. Further investigation of the field conditions revealed that stream erosion had a more significant impact on the concrete surface of the left bank, resulting in patches and holes that were not scanned by the iPhone LiDAR. As a result, the KNN algorithm used to find the nearest (2D) point cloud coordinates to the water line could not accurately represent the corresponding real-world coordinates of these locations. Figure 7 shows a box plot and scatter plot of the estimated water level for a time-lapse image captured at 13:29 on Aug 19, 2022. The patches and holes on the left bank surface caused instability in water level estimation for the region of interest. The box plot of the left bank (Cam-L-BL) was taller than that of the right bank (Cam-R-BL), indicating that the estimated water level was spread over larger values in the left bank due to the presence of these irregularities.

After analyzing the initial results, the deployable setups were modified to enhance the quality of data collection. The programming code of the Arduino device, Aava, was modified to measure five different records for water level, each time it is triggered by the camera device, Beena, and transmit the average distance to the Raspberry Pi device. This modification decreased the number of noise spikes in the measured data and allowed a better comparison between camera readings and ground truth data. The case of the camera device, Beena, was redesigned to protect the single board against rain without requiring an umbrella which makes the camera setup unstable in stormy weather and causes a decrease in the precision of measurements. Moreover, an opening is incorporated into the redesigned case to connect an external power bank to enhance the run time. Finally, the viewpoint of the camera was subtly shifted to the right to adjust the share of the river banks on the camera's field of view.

The results of the deployments on Nov 10, 2022, and Nov 11, 2022, demonstrate that modifications to the setup have significantly improved the results of the left bank (as shown in Table 3). NSE improved from approximately 0.55 for the first three setup deployments to over 0.80 for the modified deployments. Figure 8 shows the setup performances during all segments of the flood wave. The peaks were captured by the right bankline on both deployment dates, and there was no effect of noisy spikes on either camera readings or ground truth data. However, the right bank images still underestimated the water level during the rainstorms.

# 6 Conclusion

This study introduced Eye of Horus, a vision-based framework for hydrologic monitoring and measuring real-time water-related parameters, e.g., water level, from surveillance images captured during flood events. Time-lapse images and real water level correspondences were collected by Raspberry Pi camera and Arduino HC-SR05 ultrasonic sensor, respectively. Moreover, Computer Vision and Deep Learning techniques were used for semantic segmentation of water surface within the captured images and for reprojecting the 3D point cloud constructed with an iPhone LiDAR scanner, on the (2D) image plane. Eventually, the K-Nearest Neighbor algorithm was used to intersect the projected (2D) point cloud with the water line pixels extracted from the output of the Deep Learning model, to find the real-world 3D coordinates.

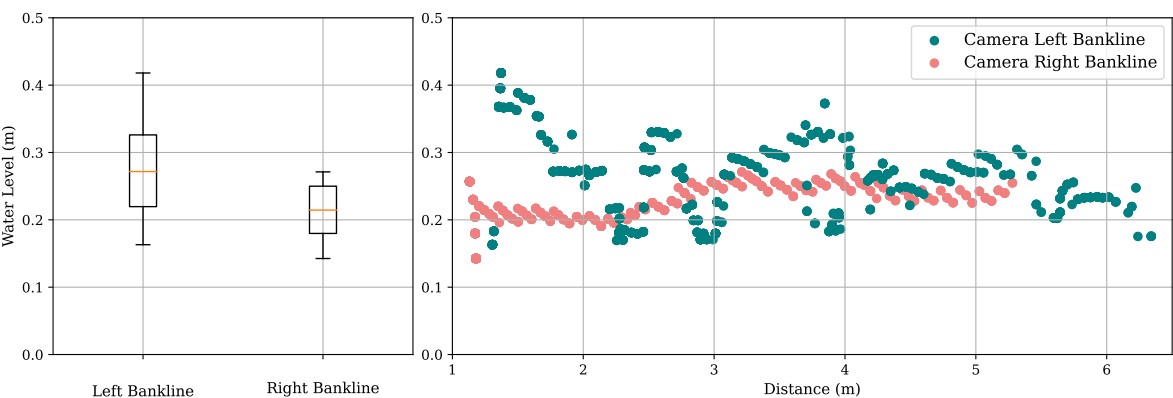

Figure 6: Scatter plot and time series plot for estimated water level by the proposed framework and measured by the ultrasonic sensor for setup deployment on (a) Aug 17, 2022 (b) Aug 19, 2022, and (c) Aug 25, 2022.

Figure 7: Water level fluctuation along both left and right banks for the flow regime for an image captured at 13:29 on Aug 19, 2022.

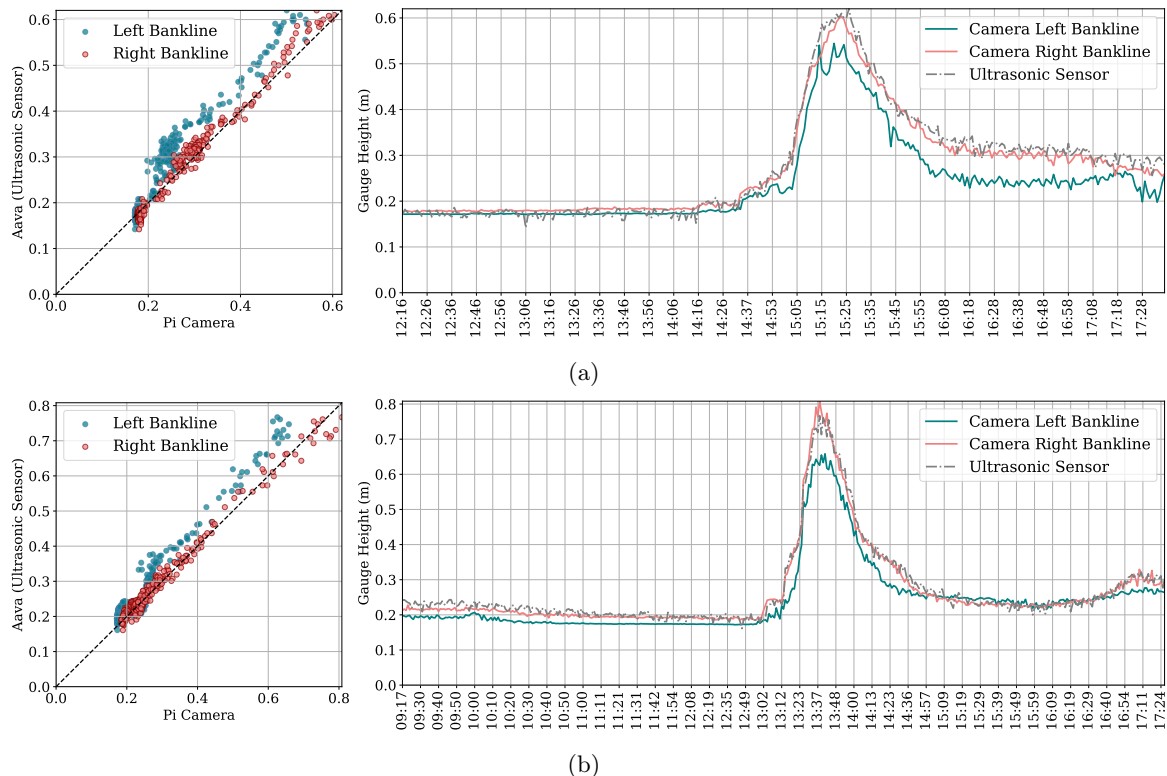

Figure 8: Scatter plot and time series plot for estimated water level by the proposed framework and measured by the ultrasonic sensor for setup deployment on (a) Nov 10, 2022, and (b) Nov 11, 2022.

A vision-based framework offers a new alternative to current hydrologic data collection and real-time monitoring systems. Hydrological models require geometric information for estimating discharge routing parameters, stage, and flood inundation maps. However, determining bankfull characteristics is a challenge due to natural or anthropogenic down-cutting of streams. Using visual sensing, stream depth, water velocity, and instantaneous streamflow at bankfull stage can be reliably measured.

# 7   Data Availability Statement

The framework and codes developed and used in this study are publicly available online in the GitHub repository (https://github.com/smhassanerfani/horus).

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
