# Peer review of "EYE OF HORUS: A VISION-BASED FRAMEWORK FOR REAL-TIME WATER LEVEL MEASUREMENT"

_EGUsphere, 2023_

## Author Comment (AC1)

Reply on RC1

Question 1: The reviewer shared his concern about the generalizability of this framework several times.

The primary aim of this study was to introduce an all-encompassing pipeline showcasing the viability of utilizing Computer Vision, Machine Learning techniques, and LiDAR data for the estimation of stream flow characteristics. This innovative framework encompasses five discrete models: a Semantic Segmentation Deep Learning (DL) model, a Camera Calibration model, a Spatial Resection model, a Projection model, and a Machine Learning (ML) model designed to index the nearest 2D projected point cloud. The authors were particularly intrigued by the prospect of synergizing these diverse models within the framework. It's worth noting that each model inherently carries its own inherent margin of error. Hence, the central inquiry revolved around whether the amalgamation of these models could still yield results that are both comparable and dependable.

The Eye of Horus presents a highly promising alternative to prevailing hydrologic data collection and real-time monitoring systems. Its applicability extends beyond mere data collection, especially in scenarios of fluvial flooding, where floods transpire as river water levels surpass the bankfull capacity and spill onto adjoining floodplains. In such circumstances, the Eye of Horus assumes a dual role, not only as an alternative data collection method but also as a source of invaluable insights for estimating elevated water thresholds in rivers and streams. Importantly, this application doesn't necessitate complete cross-sectional point cloud coverage, rendering this framework adaptable to various types of fluvial systems.

Question 2: One of the reviewer's concerns is as follows "The main one I have in mind would be to create a water segmentation network that does not rely on the images of the same camera and compare with the current results. Such datasets exist (see [1] where some of them are cited). In practice, having to annotate hundreds of images at each site to make the method work seems a hard limitation on the scope of this work."

It is crucial to mention that the core objective of this study was not centered solely on the exploration of "Transfer Learning" or delving exclusively into the technical intricacies of Deep Learning models. While these are undoubtedly pertinent components of our research, they do not stand as the primary focal points. The collaborative teams of authors involved in this study have previously contributed to the field through a spectrum of image datasets and Deep Learning models, each tailored to address intricate aspects of water and waterbody-related tasks[1][2][3].
* * *
[1] Erfani, S.M.H., Wu, Z., Wu, X., Wang, S. and Goharian, E., 2022. ATLANTIS: A benchmark for semantic segmentation of waterbody images. Environmental Modelling & Software, 149, p.105333.

[2] Erfani, S.M.H. and Goharian, E., 2022. Atex: a benchmark for image classification of water in different waterbodies using deep learning approaches. Journal of Water Resources Planning and Management, 148(11), p.04022063.

[3] Erfani, S.M.H. and Goharian, E., 2023. Vision-based texture and color analysis of waterbody images using computer vision and deep learning techniques. Journal of Hydroinformatics, 25(3), pp.835-850.

Our intention goes beyond the technical realm; we aspire to present a comprehensive and holistic approach that amalgamates various methodologies – from Computer Vision to Machine Learning and LiDAR data analysis – to advance the understanding and estimation of stream flow characteristics. In light of our collective experience in producing specialized image datasets and Deep Learning models tailored for water-centric applications, we have integrated these prior accomplishments into the foundation of our current research. By doing so, we contribute not only to the progression of individual components but also to the synthesis of a more comprehensive approach that can potentially revolutionize the field's methodologies and outcomes.

Question 3: The efficiency criteria used in the work ($R^2$, NSE, RMSE, PBIAS) should be defined with a formula for more clarity.

It has been addressed. Thank you!

Question 4: L161-170. This part should be clarified. I am not sure how the GCPs are used to register the LiDAR sub-regions that were captured. An explanation of the AruCo marker would help. Maybe simplify here and refer+merge with Section 4.3?

Ground Control Points (GCPs) have played a pivotal role as reference benchmarks in the process of aligning distinct point clouds and seamlessly amalgamating them to form a cohesive, integrated point cloud encompassing the entirety of the study area. This alignment was achieved by meticulously identifying corresponding regions within both the GCPs and the scanned point clouds, facilitating a seamless transformation of the point cloud into the coordinate system defined by the GCPs.

In the pursuit of estimating camera extrinsic parameters, ArUco markers emerged as indispensable tools. Their application necessitated recalibration of these parameters for each deployment scenario. To achieve this, the Spatial Resection model was employed—a technique demanding a minimum of three pairs of 3D-2D points to deduce the six unknowns governing rotation and translation parameters (for further insights, please refer to the authors' response to RC2, Question 4).

In the pursuit of precision, ArUco markers were meticulously installed along the creek, establishing a permanent presence. The location of the top-left corner of each marker ($X_w$, $Y_w$, $Z_w$) was meticulously surveyed using a total station, lending a high degree of accuracy to the process. To translate these real-world coordinates onto the image plane ($u$, $v$), the ArUco module within the OpenCV package was harnessed (as depicted in Figure 2).

[Figure]

*Figure 1Example of a pair of 3D-2D correspondence. (Source: docs.opencv.org).*

Question 5: Figure 1 should be better explained, especially Fig 1b with, I think, the AruCo markers & the white numbers in black background.

The incorporation of Figure 2, accompanied by an enhanced narrative in both the caption and the manuscript. This strategic enhancement aims to elevate the clarity and comprehensiveness of the visual representation, enriching the reader's understanding of the depicted content.

[Figure]

*Figure 2Visually illustrate where ArUco markers install and how they are detected.*

Question 6: L298-301/Table 2. This should be better motivated or explained. As I understand, I am not sure why non river ground truth pixels should be ignored.

When employing the Intersection over Union (IoU) metric to assess object detection or segmentation models, the omission of background pixels or regions from the calculation holds significant merit for a multitude of compelling reasons:

1. **Foreground Focus**: The paramount objective of object detection and segmentation undertakings is the precise identification and demarcation of foreground objects of interest. The deliberate exclusion of background pixels facilitates a concentrated evaluation of the IoU metric, concentrating solely on the model's proficiency in capturing and localizing these pivotal foreground entities.

2. **Localization Precision**: The IoU metric transcends mere overlap measurement between predicted and ground truth regions; it also penalizes the model for any inaccuracies in localization. The strategic omission of background pixels ensures that the model bears the brunt of penalties solely for its misidentification of object regions, sidestepping any inadvertent repercussions stemming from background misidentification.

3. **Preclusion of Inflated Scores**: The incorporation of background pixels within the IoU calculation risks generating artificially inflated scores, especially if the model proves adept at predicting background regions—potentially the predominant element within the image.

This phenomenon could potentially obscure subpar performance in the accurate detection and segmentation of the foreground objects.

4. **Consonance with Real-world Utility**: In practical, real-world contexts, the efficacy of an object detection or segmentation model hinges on its proficiency in accurately identifying and segmenting pertinent objects within a given scene. By excising background pixels, the evaluation process aligns more congruently with the intrinsic utility of the model in real-world applications.

In order to achieve accurate IoU metric computation, it is imperative to rigorously define and confine both predicted and ground truth regions to areas of substantive interest—the foreground objects. The exclusion of background pixels endows the IoU metric with enhanced significance and representativeness, offering a more insightful and authentic evaluation of the model's prowess in the domains of object detection and segmentation.

Question 7: L287-292. There is no mention of how the KNN K parameter was validated. I also wonder if that parameter played a role in the results (L376-385).

The utilization of the Customized KNN model serves as a pivotal technique for the estimation of the closest projected 2D point cloud (with k=1) to the water line pixel. This specialized model leverages the Euclidean distance computed across the image plane, enabling the pinpoint identification of the nearest point in proximity. Upon establishing this proximity, the model derives the index of the identified point, which, in turn, facilitates the retrieval of the corresponding real-world coordinates from the encompassing 3D point cloud.

Question 8: L316-331. Isn't there a possibility that the Transformer networks are "overfitting" the single camera training set?

It is pertinent to highlight that the resolution of these concerns falls outside the purview of the current study. Should the aspiration be to apply this framework across expansive domains and diverse scenarios, the authors advocate for a comprehensive approach involving extensive model training on a substantial dataset. Furthermore, the incorporation of transfer learning and fine-tuning techniques warrants careful consideration to enhance the framework's adaptability and robustness.

Question 8: Review and inclusion of sufficient references, including [1] Vandaele et al., https://doi.org/10.1007/978-3-030-71278-5_17 & [2] Vandaele et al., https://doi.org/10.5194/hess-25-4435-2021.

The references have been meticulously examined, and the authors have taken diligent care to incorporate them, along with supplementary citations, appropriately within the revised version of the manuscript.

RC2

Question 1: It should be noted somewhere (e.g. in the abstract) that the system cannot work at night.

The authors evaluated the performance of the infrared Raspberry Pi Camera Module 2 NoIR for capturing timelapse pictures during the night. The Camera Module 2 NoIR offers all the features of the standard Camera Module, but with one key difference: it lacks an infrared filter, allowing users to capture images in the dark using infrared lighting.

However, the experimental results did not yield significant findings. While we cannot definitively conclude that this system is ineffective at night, several options should be explored before reaching a final judgment. Two potential solutions are as follows:

1. Improved NoIR Cameras: It is essential to consider using higher-quality NoIR cameras, as Raspberry Pi's affordable products might not be optimal for specific purposes, such as validating vision frameworks in low-light conditions.
2. Customized Deep Learning Models for Nighttime Semantic Segmentation: Developing and utilizing deep learning models specifically designed for nighttime semantic segmentation could enhance the camera's performance.

Despite these potential improvements, the primary objective of this study was to present a comprehensive pipeline demonstrating the feasibility of using Computer Vision, Machine Learning techniques, and LiDAR data to estimate stream flow characteristics. Due to this focus, further investigation into nocturnal image capture was not pursued at this stage.

Question 2: L142 The geographical coordinates of the site would be useful.

The case study location serves as a lab scale example for testing the proposed framework and validating the results. Moreover, the study does not include hydrological analyses related to the catchment area, as it falls beyond the scope of this particular research. So, the authors have chosen not to include additional figures at this stage due to the relatively high number of figures already presented in the study.

Question 3: 4.3 L266 Is the focal length fixed or variable?

In this study we utilized Raspberry Pi Camera Module 2 which has a fixed focal length. Here is the link for official documentation for camera hardware specification. https://www.raspberrypi.com/documentation/accessories/camera.html

Question 4: Why don't you estimate the DLT parameters directly? What is the advantage of estimating the intrinsic parameters beforehand?

The DLT (Direct Linear Transformation) method requires a minimum of 6 pairs of known 3D-2D correspondences to establish 12 equations and estimate all 12 parameters of the projection matrix. This matrix encompasses both the intrinsic and extrinsic parameters of the camera. The intrinsic parameters remain constant for a specific camera model (such as the Raspberry Pi Camera Module 2) and can be reused for all images captured by that camera. However, the extrinsic parameters change whenever the camera's location is altered. Consequently, for each setup deployment, recalculation of the extrinsic parameters is necessary to reconstruct the projection matrix.

To reduce the number of required 3D-2D correspondence pairs, we employed the Spatial Resection method. This algorithm leverages the existing intrinsic parameters and focuses on recalculating only the extrinsic parameters. With a minimum of 3 pairs of 3D-2D points, the Spatial Resection method can estimate the 6 unknowns, which represent the rotation and translation parameters (Figure 1).

**Direct Linear Transform**

**Projection matrix P₃ₓ₄**

$$\lambda \begin{bmatrix} u^{(I)} \\ v^{(I)} \\ 1 \end{bmatrix} = \begin{bmatrix} f_u \mathbf{r}_1^t + u_0 \mathbf{r}_3^t & f_u t_X + u_0 t_z \\ f_v \mathbf{r}_2^t + v_0 \mathbf{r}_3^t & f_v t_Y + v_0 t_z \\ \mathbf{r}_3^t & t_z \end{bmatrix} \begin{bmatrix} X^{(W)} \\ Y^{(W)} \\ Z^{(W)} \\ 1 \end{bmatrix}$$

Source: Slides of Computer Vision Course by Dr. Yan Tong
(Professor of Department of Computer Science & Engineering, University of South Carolina)

**Spatial Resection**
Projective 3 Points

$$\mathbf{x} = KR[I_3| - X_O]\mathbf{X}$$

observed image point

$$\begin{bmatrix} f_u & \gamma & u_0 \\ 0 & f_v & v_0 \\ 0 & 0 & 1 \end{bmatrix}$$

**3 rotations**

**3 translations**

control point coordinates (given)

*Figure 3Direct Linear Transform method needs at least 6 pairs of known 3D-2D correspondence to calculate the projection matrix while Spatial Resection needs 3 pairs of 3D-2D points.*

Editorial:
Question 5: Eq. 1: camera (camera)

It's addressed. Thank you!

---

## Author Comment (AC2)

Reply on RC2

Question 1: It should be noted somewhere (e.g. in the abstract) that the system cannot work at night.

The performance assessment of the infrared Raspberry Pi Camera Module 2 NoIR for capturing nighttime timelapse images was conducted by the authors. The Camera Module 2 NoIR shares all the features of the standard Camera Module, except for one distinctive attribute: the absence of an infrared filter, enabling the acquisition of images in low-light conditions using infrared illumination.

However, the experimental outcomes yielded inconclusive results. While definitive assertions regarding the system's nocturnal efficacy remain elusive, prudent exploration of alternative avenues is warranted before arriving at a final verdict. The following two potential courses of action merit consideration:

1. **Enhanced NoIR Cameras**: It is prudent to contemplate the utilization of superior quality NoIR cameras, particularly since Raspberry Pi's cost-effective offerings may not align optimally with specific requirements such as validating vision frameworks under low-light circumstances.

2. **Tailored Deep Learning Models for Nighttime Semantic Segmentation**: The development and application of deep learning models expressly tailored for nighttime semantic segmentation could conceivably bolster the camera's performance in challenging lighting conditions.

Notwithstanding these prospective enhancements, it is pivotal to underscore that the principal focus of this study was to introduce a holistic framework showcasing the viability of employing Computer Vision, Machine Learning techniques, and LiDAR data for the estimation of stream flow characteristics. Given this concentrated objective, the pursuit of further exploration into nocturnal image capture was regrettably deferred at this juncture.

Question 2: L142 The geographical coordinates of the site would be useful.

The selected case study location functions as a small-scale illustration, providing a testing ground for the proposed framework's assessment and result validation. It is important to note that the scope of this research does not encompass hydrological analyses pertaining to the catchment area. Consequently, the decision has been made to abstain from incorporating supplementary figures at this juncture, given the already substantial number of figures already presented in the study.

Question 3: 4.3 L266 Is the focal length fixed or variable?

Within the confines of this study, the Raspberry Pi Camera Module 2 was harnessed, notable for its unalterable focal length. The official documentation detailing the hardware specifications of this camera can be accessed via the following link: https://www.raspberrypi.com/documentation/accessories/camera.html.

Question 4: Why don't you estimate the DLT parameters directly? What is the advantage of estimating the intrinsic parameters beforehand?

The application of the Direct Linear Transformation (DLT) method mandates a minimum of 6 pairs of established 3D-2D correspondences. This prerequisite yields 12 equations, enabling the estimation of the complete set of 12 parameters inherent to the projection matrix. This matrix encapsulates both the camera's intrinsic and extrinsic parameters. Notably, intrinsic parameters remain invariant for a given camera model (e.g., Raspberry Pi Camera Module 2), enabling their

**Direct Linear Transform**

Projection matrix $P_{3x4}$

$$\lambda \begin{bmatrix} u^{(I)} \\ v^{(I)} \\ 1 \end{bmatrix} = \begin{bmatrix} f_u \mathbf{r}_1^t + u_0 \mathbf{r}_3^t & f_u t_X + u_0 t_z \\ f_v \mathbf{r}_2^t + v_0 \mathbf{r}_3^t & f_v t_Y + v_0 t_z \\ \mathbf{r}_3^t & t_z \end{bmatrix} \begin{bmatrix} X^{(W)} \\ Y^{(W)} \\ Z^{(W)} \\ 1 \end{bmatrix}$$

Source: Slides of Computer Vision Course by Dr. Yan Tong
(Professor of Department of Computer Science & Engineering,
University of South Carolina)

**Spatial Resection**
**Projective 3 Points**

[Figure]

$$\mathbf{x} = KR[I_3| - X_O]\mathbf{X}$$

observed image point

$$\begin{bmatrix} f_u & \gamma & u_0 \\ 0 & f_v & v_0 \\ 0 & 0 & 1 \end{bmatrix}$$

control point coordinates (given)

**3 translations**

**3 rotations**

*Figure 1. Direct Linear Transform method needs at least 6 pairs of known 3D-2D correspondence to calculate the projection matrix while Spatial Resection needs 3 pairs of 3D-2D points.*

reuse across all images captured by that specific camera. Conversely, the extrinsic parameters undergo variation with shifts in camera position. Consequently, recalculating extrinsic parameters becomes imperative for each deployment scenario, ensuring the accurate reconstruction of the projection matrix.

To streamline the requirement for 3D-2D correspondence pairs, the Spatial Resection method was introduced. This algorithm adeptly capitalizes on existing intrinsic parameters, directing its focus toward the reevaluation of solely extrinsic parameters. By employing a mere minimum of 3 pairs of 3D-2D points, the Spatial Resection method successfully deduces the 6 unknowns essential for representing rotation and translation parameters (as illustrated in Figure 1).

Editorial:
Question 5: Eq. 1: camera (camera)

It has been fixed. Thank you!